# Dietary fibre and whole grains in diabetes management: Systematic review and meta-analyses

**Andrew N. Reynolds**[1,2]*, **Ashley P. Akerman**[1,3], **Jim Mann**[1,2]

**1** Department of Medicine, University of Otago, Dunedin, Otago, New Zealand, **2** Edgar National Centre for Diabetes and Obesity Research, University of Otago, Dunedin, New Zealand, **3** School of Physical Education, Sports, and Exercise Science, University of Otago, Dunedin, New Zealand

* andrew.reynolds@otago.ac.nz

**Data Availability Statement:** All relevant data may be obtained within the manuscript, the supporting information file, or from the original articles.

**Funding:** This work was funded by the Department of Medicine at the University of Otago, New

## Abstract

### Background

Fibre is promoted as part of a healthy dietary pattern and in diabetes management. We have considered the role of high-fibre diets on mortality and increasing fibre intake on glycaemic control and other cardiometabolic risk factors of adults with prediabetes or diabetes.

### Methods and findings

We conducted a systematic review of published literature to identify prospective studies or controlled trials that have examined the effects of a higher fibre intake without additional dietary or other lifestyle modification in adults with prediabetes, gestational diabetes, type 1 diabetes, and type 2 diabetes. Meta-analyses were undertaken to determine the effects of higher fibre intake on all-cause and cardiovascular mortality and increasing fibre intake on glycaemic control and a range of cardiometabolic risk factors. For trials, meta regression analyses identified further variables that influenced the pooled findings. Dose response testing was undertaken; Grading of Recommendations Assessment, Development and Evaluation (GRADE) protocols were followed to assess the quality of evidence. Two multicountry cohorts of 8,300 adults with type 1 or type 2 diabetes followed on average for 8.8 years and 42 trials including 1,789 adults with prediabetes, type 1, or type 2 diabetes were identified. Prospective cohort data indicate an absolute reduction of 14 fewer deaths (95% confidence interval (CI) 4–19) per 1,000 participants over the study duration, when comparing a daily dietary fibre intake of 35 g with the average intake of 19 g, with a clear dose response relationship apparent. Increased fibre intakes reduced glycated haemoglobin (HbA1c; mean difference [MD] −2.00 mmol/mol, 95% CI −3.30 to −0.71 from 33 trials), fasting plasma glucose (MD −0.56 mmol/L, 95% CI −0.73 to −0.38 from 34 trials), insulin (standardised mean difference [SMD] −2.03, 95% CI −2.92 to −1.13 from 19 trials), homeostatic model assessment of insulin resistance (HOMA IR; MD −1.24 mg/dL, 95% CI −1.72 to −0.76 from 9 trials), total cholesterol (MD −0.34 mmol/L, 95% CI −0.46 to −0.22 from 27 trials), low-density lipoprotein (LDL) cholesterol (MD −0.17 mmol/L, 95% CI −0.27 to −0.08 from 21 trials), triglycerides (MD −0.16 mmol/L, 95% CI −0.23 to −0.09 from 28 trials), body weight (MD −0.56 kg,

Zealand (ANR, APA), the Healthier Lives National Science Challenge, New Zealand (JM), and the Edgar Diabetes and Obesity Research Centre, New Zealand (ANR, JM). The funders had no role in study design, data collection and analysis, decision to publish, or preparation of the manuscript.

**Competing interests:** The authors have declared that no competing interests exist.

**Abbreviations:** BMI, body mass index; CI, confidence interval; CRP, C-reactive protein; CVD, cardiovascular disease; DNSG, Diabetes and Nutrition Study Group of the European Association for the Study of Diabetes; FPG, fasting plasma glucose; GRADE, grading of recommendations assessment, development and evaluation; HbA1c, glycated haemoglobin; HDL, high-density lipoprotein; HOMA IR, homeostatic model assessment of insulin resistance; LDL, low-density lipoprotein; MD, mean difference; PICO, Population, Intervention, Control, and Outcomes; RR, relative risk; SMD, standardised mean difference.

95% CI −0.98 to −0.13 from 18 trials), Body Mass Index (BMI; MD −0.36, 95% CI −0·55 to −0·16 from 14 trials), and C-reactive protein (SMD −2.80, 95% CI −4.52 to −1.09 from 7 trials) when compared with lower fibre diets. All trial analyses were subject to high heterogeneity. Key variables beyond increasing fibre intake were the fibre intake at baseline, the global region where the trials were conducted, and participant inclusion criteria other than diabetes type. Potential limitations were the lack of prospective cohort data in non-European countries and the lack of long-term (12 months or greater) controlled trials of increasing fibre intakes in adults with diabetes.

## Conclusions

Higher-fibre diets are an important component of diabetes management, resulting in improvements in measures of glycaemic control, blood lipids, body weight, and inflammation, as well as a reduction in premature mortality. These benefits were not confined to any fibre type or to any type of diabetes and were apparent across the range of intakes, although greater improvements in glycaemic control were observed for those moving from low to moderate or high intakes. Based on these findings, increasing daily fibre intake by 15 g or to 35 g might be a reasonable target that would be expected to reduce risk of premature mortality in adults with diabetes.

---

## Author summary

### Why was the study done?

- Higher intakes of dietary fibre are associated with a reduction in premature mortality and incidence of a wide range of noncommunicable diseases and their risk factors in the population at large. We have considered the role of high-fibre diets on mortality and increasing fibre intake on cardiometabolic risk factors of adults with prediabetes or diabetes.

- We conducted a systematic review and meta-analyses to inform an update of European nutrition guidelines for diabetes management.

### What did the researchers do and find?

- We examined prospective cohort studies of dietary fibre intake and mortality as well as controlled trials, which considered the effects of increasing fibre intakes on glycaemic control and risk factors for cardiovascular disease. We explored dose response relationships between dietary fibre and these outcomes to determine a more reliable estimate for recommended intakes.

- We found that participants in prospective cohort studies consuming higher intakes of dietary fibre had a reduced risk of premature mortality when compared with those with lower fibre intakes.

- We also found that the results from controlled trials were complementary, with increasing intake of fibre improving glycaemic control and other risk factors for cardiovascular disease, such as cholesterol levels and body weight.

### What do the findings mean?

- Those with prediabetes, type 1, or type 2 diabetes should increase their dietary fibre intakes by 15 g per day or to 35 g per day.

- One practical way to increase fibre intakes is to replace refined grain products with whole grain foods.

## Introduction

The observation derived from uncontrolled studies in the 1970s [1–3] that amongst people with diabetes glycaemic control improved when dietary carbohydrate was increased, led to the resurrection of an earlier suggestion [4] that a relatively high carbohydrate intake might be preferable to what was then the cornerstone of dietary advice for diabetes, a low (typically around 40% of total energy) carbohydrate diet [5]. Connor and Connor reached a similar conclusion based on the increased risk of coronary heart disease in people with diabetes and the adverse effects on blood lipids of a low carbohydrate diet, which was typically high in saturated fat [6].

Subsequent controlled trials confirmed the potential of relatively high- compared with low-carbohydrate intake to reduce levels of glycated haemoglobin, improve diurnal blood glucose profiles and levels of total and low-density lipoprotein (LDL) cholesterol in people with diabetes [7–10]. However, these benefits were only apparent when the carbohydrate-containing foods were rich in dietary fibre [11,12]. Other randomised trials and a 2013 systematic review and meta-analysis of trials demonstrated in people with type 2 diabetes the benefit of dietary fibre on glycaemic control when total carbohydrate intake remained constant [13,14]. There have been 2 more recent reviews, one related to the effects of viscous fibre [15], and the other was an umbrella review that did not provide any new analyses [16].

We have undertaken a systematic review and meta-analysis of dietary fibre in diabetes management. The inclusion of more trials and consideration of prospective cohort studies with additional information obtained from the authors has enabled us to extend earlier observations and examine several additional issues relevant to nutrition recommendations for adults with diabetes. We have explored whether dietary fibre also has the potential to improve glycaemic control in type 1 diabetes and prediabetes and favourably influence a range of cardiometabolic risk factors in addition to glycaemic control (cholesterol [total, LDL, high-density lipoprotein (HDL)], triglycerides, body weight, body mass index, waist circumference, fasting insulin, homeostatic model assessment of insulin resistance [HOMA IR], blood pressure, and C-reactive protein [CRP]). Examination of prospective data has enabled us to examine whether these benefits translate into a reduction in morbidity and mortality. Our expanded exploration of existing trials as well as cohort studies has permitted us to also examine, for the first time to our knowledge, whether dose response relationships exist between dietary fibre and a range of clinically relevant outcome measures and to provide a justification for a quantitative recommendation for fibre intakes in diabetes management.

## Methods

We followed PRISMA reporting standards for systematic reviews and meta-analyses [17]. The protocol for this systematic review was prospectively registered CRD42018089162.

### PICO tables and eligibility criteria

The overall objectives of this systematic review and meta analysis were to determine whether higher fibre intake has the potential to reduce risk of premature mortality, or whether increasing fibre intake, regardless of source, improve glycaemic control or cardiometabolic risk factors in all types of diabetes regardless of treatment regimen. The PICO (Population, Intervention, Control, and Outcomes) tables (S1 Appendix), agreed by a subcommittee of the Diabetes and Nutrition Study Group of the European Association for the Study of Diabetes (DNSG), were developed to address the research question 'what is the role of high fibre diets in diabetes management'. Prospective cohorts or studies nested in cohorts of adults with prediabetes or impaired glucose tolerance, gestational diabetes, or type 1 or type 2 diabetes that reported all-cause or cardiovascular mortality were considered eligible. Controlled trials were required to report on glycaemic control, insulin measures, blood lipids, adiposity, blood pressure, or CRP. Parallel and crossover trials of at least 6 weeks duration in which the intervention was an increase in whole grains or dietary fibre [18] were included. Six weeks was considered by the DNSG subcommittee to be the minimum trial duration in which HbA1c (the primary outcome) may be expected to change. Adults identified as having prediabetes or impaired glucose tolerance, gestational diabetes, or type 1 or type 2 diabetes, regardless of type of medication and presence of comorbidities were eligible. Eligible trials included studies in which foods were provided or those in which dietary advice regarding fibre and whole grain was given with no further instruction to change intake of macronutrients or energy. Trials comparing equivalent amounts of one fibre type with another were not included.

### Literature search

Eligible controlled trials and prospective cohort studies were identified in the same search, using Cochrane search strategies [19]. OVID Medline, Embase, PubMed, and the Cochrane Central Register of Controlled Trials were searched up to 18 January 2019. These searches included a range of terms for diabetes or impaired glucose metabolism, dietary carbohydrate and fibre, and trial design (available in full in S2 Appendix). This online strategy was augmented by hand searches of reference lists to identify other potentially eligible publications. Commercially available software was used to remove duplicates and aid screening [20]. No date or language restrictions were applied.

### Study selection, data extraction, and risk of bias assessment

Literature searches, identification of eligible studies, data extraction, and bias assessment were undertaken independently by at least 2 researchers, with any discrepancies resolved with an additional reviewer. Data were extracted using pretested forms [21]. For prospective cohort studies, the most adjusted values for effect size were extracted. For controlled trials, we used baseline data and those from the last time point except for in sensitivity analyses of trial duration in which we included data from any time point from 6-weeks trial duration. We used the Newcastle-Ottawa Scale [22] to assess risk of bias of prospective cohort studies and the Cochrane Collaboration's tool for assessing risk of bias in randomised trials [23].

## Data analysis

For prospective cohort studies, we received additional data from the authors, such as country-specific effect size estimates. We considered the relationship between fibre intake and all-cause or cardiovascular disease related mortality by comparing the highest intake quantile with the lowest intake quantile [24]. Dose response relationships were considered with restricted cubic splines in a two-stage, random effects model [25] after testing for linearity. We measured the risk reduction in all-cause and cardiovascular mortality on the plotted line between an average intake of 19 g per day [26] and the highest intake for which data were available.

For controlled trials, we analysed the mean difference between intervention and control groups with generic inverse variance models and random effects. We used the mean difference between change from baseline values when reported or calculated them with the reported data. For trials with more than one eligible intervention, we have avoided a unit of analysis error by splitting the size of the control group accordingly [19]. For outcomes reported in unique units, we report the standardised mean difference. Correlation coefficients were obtained from publications when reported or imputed from the average of available correlation coefficients per outcome.

The initial calculations for each trial outcome involved pooled data from all eligible studies. However, given the broad nature of the research question, additional factors influencing the initial findings could not be discounted. We considered these factors with a series of univariate meta regression analyses to identify variables beyond the increase in fibre intakes. Variables considered for each outcome were specified in the PICO table as potential sources of bias, trial characteristics, participant characteristics such as diabetes type, or intervention parameters such as the type of fibre prescribed. We ran additional meta regression analyses beyond those stated in the PICO table to consider the generalisability of our results for patients with comorbidities. For trials, we used meta regression analyses with restricted cubic splines of the measured (preferred) or prescribed fibre increase, when available, as dose response testing. When we identified an interaction with baseline fibre intake, we tested the fibre increase as a percentage of the baseline fibre intake.

For both prospective cohort studies and controlled trials, heterogeneity was assessed with the $I^2$ statistic [27] and the Cochrane Q test [28]. Sensitivity analyses were conducted when an $I^2$ statistic was found to be more than 50% or a $p$ for heterogeneity of <0.10. The effect of each individual study on the pooled result was considered with an influence analysis. This involved the removal of intervention data from the pooled estimate one at a time. Small study effects, as might be seen with publication bias, were assessed with Egger's test [29] and the trim and fill method [30]. Analyses were performed in Stata statistical software [31] or the 'metafor' package of R [32]. The most relevant analyses are presented in the results section and the full analyses in the supplementary material (S3–S17 Appendices). We used Grading of Recommendations Assessment, Development and Evaluation (GRADE) protocols [33] to calculate absolute risk reductions and evaluate the quality of the body of evidence as either high, moderate, low, or very low. Quality of the evidence was assessed by the research team and revised if required after discussion with the DNSG.

## Results

A flow chart of identified studies is shown in Fig 1. Data from 2 multicountry prospective cohort studies including 8,300 adults with type 1 or type 2 diabetes residing in 22 countries followed for a mean duration of 8.8 years [34,35] and 42 controlled trials with 1,789 participants were included in the meta-analyses [36–76]. Trial size ranged from 8 to 185 participants, with interventions increasing fibre intakes by 1 to 45 g per day. The vast majority of interventions

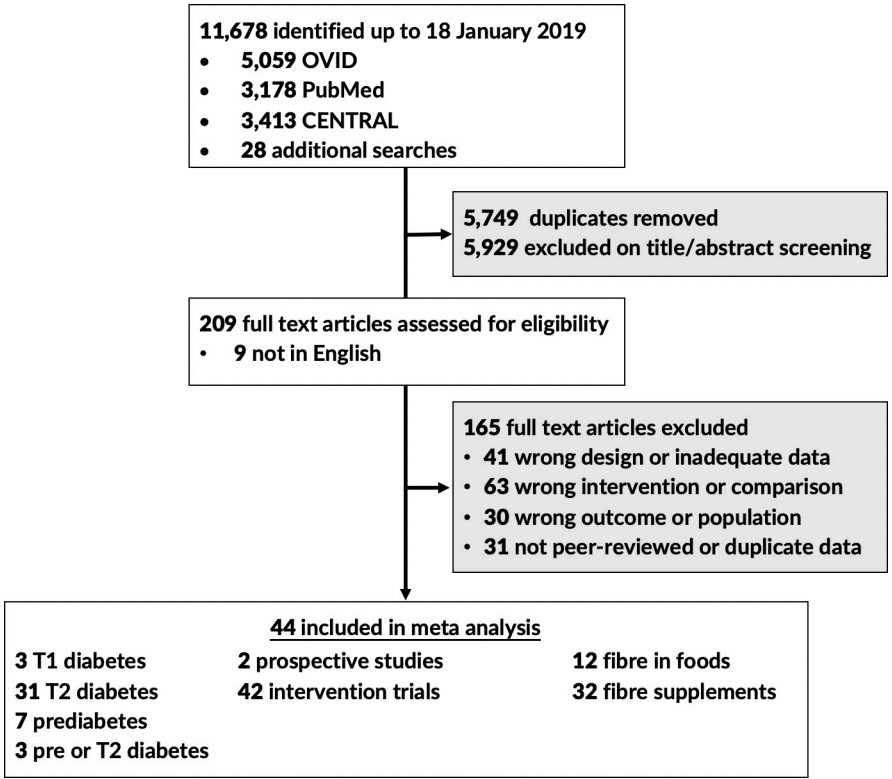

**11,678** identified up to 18 January 2019
- **5,059** OVID
- **3,178** PubMed
- **3,413** CENTRAL
- **28** additional searches

**5,749** duplicates removed
**5,929** excluded on title/abstract screening

**209** full text articles assessed for eligibility
- **9** not in English

**165** full text articles excluded
- **41** wrong design or inadequate data
- **63** wrong intervention or comparison
- **30** wrong outcome or population
- **31** not peer-reviewed or duplicate data

**44 included in meta analysis**

| | | |
|---|---|---|
| **3** T1 diabetes | **2** prospective studies | **12** fibre in foods |
| **31** T2 diabetes | **42** intervention trials | **32** fibre supplements |
| **7** prediabetes | | |
| **3** pre or T2 diabetes | | |

**Fig 1. Flow chart indicating the process by which eligible prospective cohort studies and controlled trials were identified.**

were 6 to 12 weeks long (93%), with only 1 trial undertaken for a full year. No eligible trials of women with gestational diabetes were identified. A description of identified prospective cohort studies and controlled trials are available in the supplementary materials (S2 Appendix).

## Mortality from prospective cohort studies

All-cause mortality was appreciably reduced when comparing the highest fibre intakes with the lowest (relative risk (RR) 0.55, 95% CI 0.35–0.86, $I^2$ 0%) over a weighted mean duration of 8.8 years. The estimated cardiovascular mortality risk reduction (RR 0.61, 95% CI 0.26–1.42, $I^2$ 10%) was comparable ($p = 0.811$), however confidence intervals were wide because of the limited number of outcome events available. Influence analysis did not identify any study or country specific data that significantly differed from the pooled result, and Egger's tests did not suggest publication bias. The relationship between fibre intake and all-cause and cardiovascular mortality in adults with type 1 and type 2 diabetes using country level data when possible is shown in Fig 2. There was a 35% (95% CI 10%–48%) risk reduction in all-cause mortality associated with an intake of 35 g per day compared with 19 g per day. This resulted in an absolute risk reduction of 14 (95% CI 4–19) fewer deaths per 1,000 participants over the duration of the studies included. The plotted relationship was not linear ($p = 0.005$). Further details are shown in the supplementary material (S3 Appendix).

## Glycaemic control

Thirty-three controlled trials of increasing fibre intakes on glycated haemoglobin (HbA1c) are shown in Fig 3. When considering all eligible trials, regardless of the amount of fibre

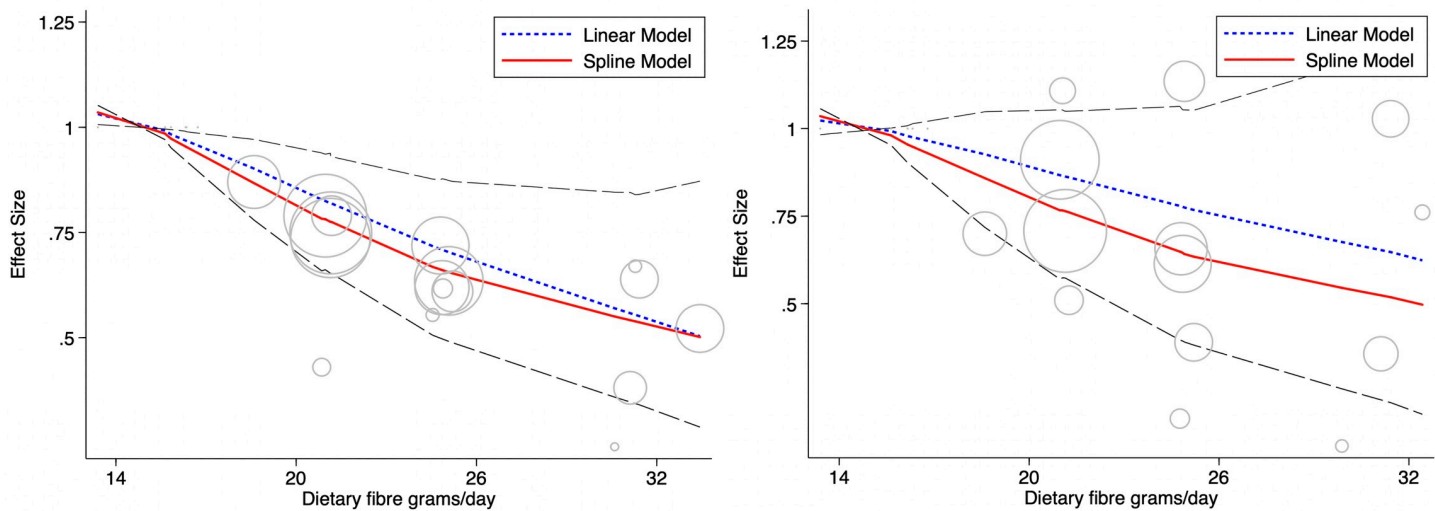

**Fig 2. Dietary fibre intake and all-cause (left) and CVD mortality (right) in cohorts with type 1 or type 2 diabetes.** The 95% CIs are shown as dashed lines. Cohort data are presented as country specific estimates when possible. Circle size indicates the weighting of each data point, with bigger circles indicating greater influence. CVD, cardiovascular disease.

prescribed, HbA1c improved with the interventions when compared with the controls (MD −2.00 mmol/mol, 95% CI −3.30 to −0.71) as shown in Table 1. Fasting glucose levels also improved with increased fibre intake (MD −0.56, 95% CI −0.73 to −0.38 from 34 trials). Influence analysis did not identify any trials that significantly influenced the pooled result for HbA1c or fasting blood glucose. There was no evidence of publication bias in the results for HbA1c and fasting blood glucose. Meta regression did not identify differences in the pooled results for HbA1c or for fasting plasma glucose because of diabetes type, diabetes medication, trial risk of bias, fibre type, trial size, or trial duration. Meta regression did indicate the reduction in HbA1c was greater in trials that did not control the weight of participants during the trials (MD −2.67 mmol/mol, 95% CI −4.18 to −1.16 from 28 trials) when compared with trials that did (MD 1.26 mmol/mol, 95% CI −0.15 to 2.68 from 5 trials). Meta regression also indicate the reduction in fasting plasma glucose was greater in trials that included participants with comorbidities (MD −0.91 mmol/L, 95% CI −1.46 to −0.36 from 15 trials) than trials that excluded participants with comorbidities (MD −0.26, 95% CI −0.46 to −0.05 from 19 trials). The results for fasting plasma glucose varied by the global region that the trial was conducted in. Further analyses for HbA1c and fasting blood glucose are shown in the supplementary material (S4 and S5 Appendices). Dose response testing for HbA1c and fasting glucose are shown in Fig 4. Both HbA1c and fasting glucose curves are plotted as a percentage of the baseline fibre intake because there was an interaction with baseline fibre intake when reported, indicating greater benefits were observed for those moving from low to moderate or high intakes.

## Blood lipids

Twenty-seven controlled trials of increasing fibre intakes on total cholesterol are shown in Fig 5. The summary effects of increasing fibre intake on the reported markers of cholesterol and triglycerides are shown in Table 1. Total cholesterol, LDL cholesterol, and triglycerides all reduced with increased fibre intakes. Influence analysis did not identify any trials that significantly influenced the pooled result for total cholesterol, LDL cholesterol, HDL cholesterol, and

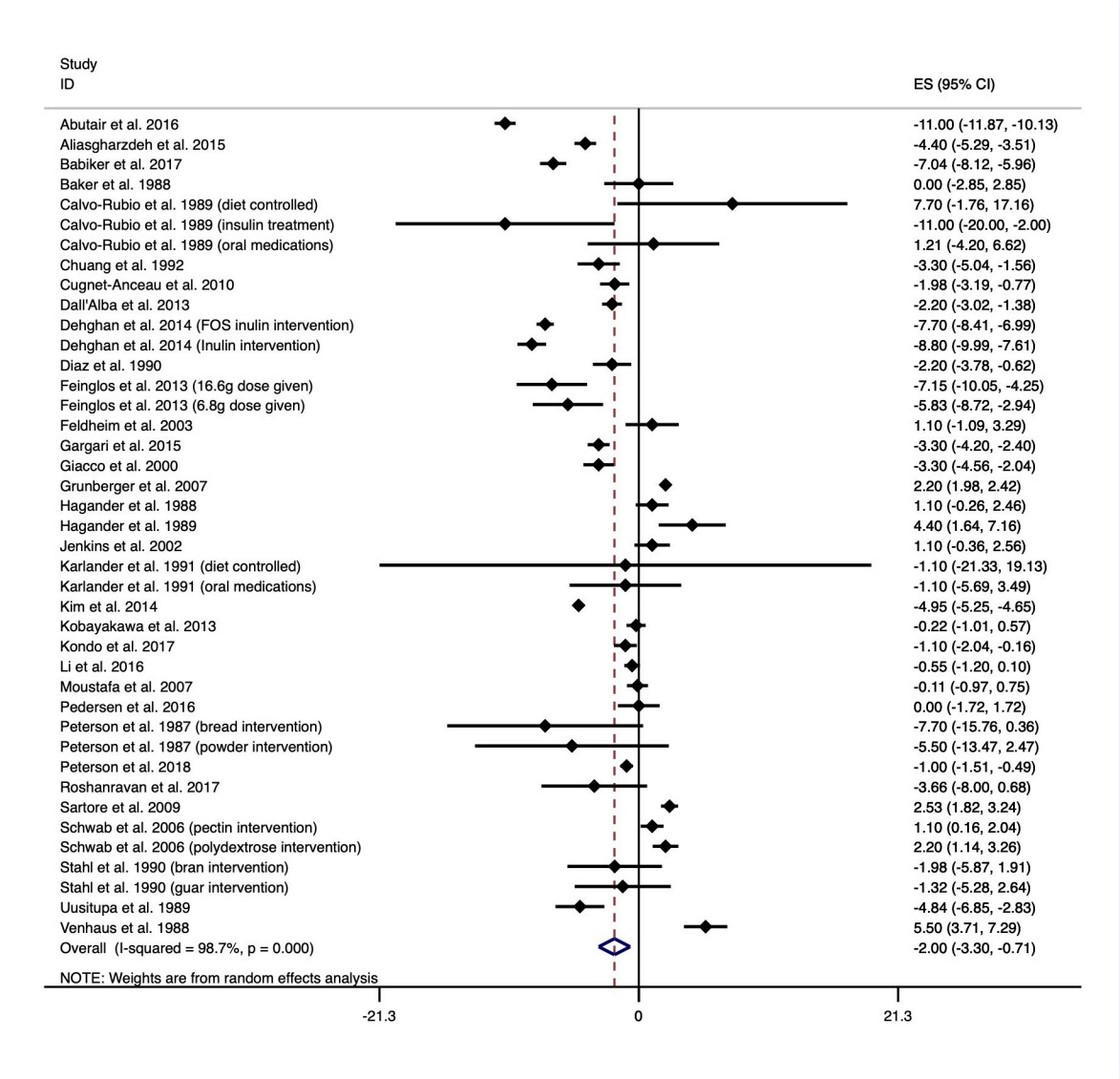

**Fig 3. Mean difference in glycated haemoglobin (mmol/mol) between intervention and control groups from trials of increasing fibre intakes.** Negative values show a decrease in HbA1c when increasing fibre intakes (the intervention). Intervention description is shown in brackets next to the first author's name and year published in which a trial reported on multiple eligible intervention arms. Medication regimen is also shown in brackets in which the results were reported separately and not as a mean result. ES, effect size; FOS, fructooligosaccharides.

triglycerides. No publication bias was observed for these outcomes. Meta regression analyses identified the pooled results for the increasing fibre intakes, and the reduction in total cholesterol, LDL cholesterol, and triglycerides were robust with no differences due to diabetes type, diabetes medication, trial risk of bias, fibre type, trial size, or trial duration. Meta regression indicated that the results for HDL cholesterol varied by global region where the trials were conducted and were higher in trials of nonviscous fibres (MD 0.11 mmol/L, 95% CI 0.03–0.18 from 6 trials) than in trials of viscous fibres (MD 0.02 mmol/L, 95% CI −0.01 to 0.04 from 11 trials). Dose response testing for total cholesterol indicated the improvement is dose dependent, with higher intakes leading to greater improvements (shown in Fig 4). Further analyses for fibre intakes and blood lipids are shown in the supplementary material (S6–S9 Appendices).

**Table 1. Summary differences in glycaemic control and cardiometabolic risk factors between intervention and control groups from trials of increasing fibre intakes.**

| Outcome | Trials | Participants (I/C) | Initial $I^2$ | MD (95% CI) |
|---|---|---|---|---|
| HbA1c (mmol/mol) | 33 | 815/738 | 98.7% | −2.00 (−3.30 to −0.71) |
| Fasting plasma glucose (mmol/L) | 34 | 936/871 | 99.1% | −0.56 (−0.73 to −0.38) |
| Total cholesterol (mmol/L) | 27 | 662/605 | 98.0% | −0.34 (−0.46 to −0.22) |
| LDL cholesterol (mmol/L) | 21 | 559/512 | 96.2% | −0.17 (−0.27 to −0.08) |
| HDL cholesterol (mmol/L) | 25 | 722/666 | 96.7% | 0.04 (0.01–0.07) |
| Triglycerides (mmol/L) | 28 | 760/708 | 97.7% | −0.16 (−0.23 to −0.09) |
| Body weight (kg) | 18 | 455/422 | 98.2% | −0.56 (−0.98 to −0.13) |
| BMI (kg/m$^2$) | 14 | 382/381 | 97.4% | −0.36 (−0.55 to −0.16) |
| Waist circumference (cm) | 8 | 178/171 | 97.4% | −1.42 (−2.63 to −0.21) |
| Systolic blood pressure (mmHg) | 12 | 325/295 | 98.6% | −1.86 (−4.85 to 1.12) |
| Diastolic blood pressure (mmHg) | 12 | 325/295 | 97.1% | −1.19 (−2.87 to 0.49) |
| C-reactive protein (SMD) | 7 | 216/217 | 96.9% | −2.80 (−4.52 to −1.09) |
| Fasting plasma insulin (SMD) | 19 | 489/458 | 96.4% | −2.03 (−2.92 to −1.13) |
| HOMA IR (mg/dL) | 9 | 292/289 | 99.7% | −1.24 (−1.72 to −0.76) |

**Abbreviations:** BMI, Body Mass Index; HDL, high-density lipoprotein; HOMA IR, homeostatic model assessment of insulin resistance; I/C, intervention/control; LDL, low-density lipoprotein; SMD, standardised mean difference

### Body weight and additional cardiometabolic risk factors

Pooled estimates for the relationship between increasing dietary fibre and markers of adiposity and additional cardiometabolic risk factors are shown in Table 1. Increasing fibre intake reduced body weight (MD −0.56 kg, 95% CI −0.98 to −0.13 from 18 trials), BMI (MD −0.36 kg/m$^2$, 95% CI −0.55 to −0.16 from 14 trials), and waist circumference (MD −1.42 cm, 95% CI −2.63 to −0.21 from 8 trials), despite no advice to reduce energy intake. For waist circumference, removal of 2 trials [47, 74] in influence analysis did not remove the significance of the pooled effect (MD −1.59, 95% CI −3.07 to 0.10 cm). No publication bias was observed for these outcomes. Meta regression indicated the results for body weight and waist circumference were robust, with no differences due to diabetes type, diabetes medication, trial risk of bias, fibre type, trial size, or trial duration. BMI results differed by baseline fibre intake. Dose response testing for body weight is shown in Fig 4. There was no evidence of dose response associations between increasing fibre intake and measures of body weight. Further analyses for these outcomes are shown in the supplementary materials (S10–S12 Appendices). Fasting plasma insulin, C-reactive protein levels, and HOMA IR improved with higher fibre diets; however, blood pressure did not (as shown in Table 1). Influence analysis did not identify any trial that significantly influenced the pooled results. There was a probable publication bias for the evidence relating to fasting plasma insulin, although trim and fill analyses did not appreciably change the observed effect size (SMD −2.48, 95% CI −3.56 to −1.42). Meta regression indicated the results for systolic and blood pressure were robust, with no differences due to diabetes type, diabetes medication, trial risk of bias, fibre type, trial size, or trial duration. Results indicated that for the outcome insulin, dietary fibre trials conducted in the Middle East produced a greater reduction (SMD −6.22, 95% CI −10.84 to −1.61) than trials in Asia (SMD −2.14, 95% CI −3.48 to −0.80) or Europe (SMD −0.65, 95% CI −1.71 to 0.41). Further analyses and information on these outcomes are available in the supplementary material (S13–S17 Appendices).

## Heterogeneity

Pooled data from the 2 multicountry prospective cohort studies were largely homogenous, with an $I^2 \leq 10\%$. Conversely, pooled controlled trial data had high heterogeneity for each outcome assessed ($I^2 \geq 90\%$). Potential sources of bias or markers of trial quality were not identified by meta regression analyses as contributors to heterogeneity nor was the type of diabetes. All subgroups of variables identified as relevant by meta regression analyses are shown in the GRADE tables of the supplementary materials (S18 Appendix). Small study effects, influence analysis, and meta regression analyses each failed to identify a single or consistent cause for the high heterogeneity within the pooled data from trials; however, they did exclude multiple sources of bias or systematic error as determinants of the presented results.

## Risk of bias assessment and GRADE tables

Risk of bias was low in the prospective cohort studies, which both achieved 8 out of a possible 9 on the Newcastle-Ottawa Score. The trials provided sufficient reporting for the assessment of

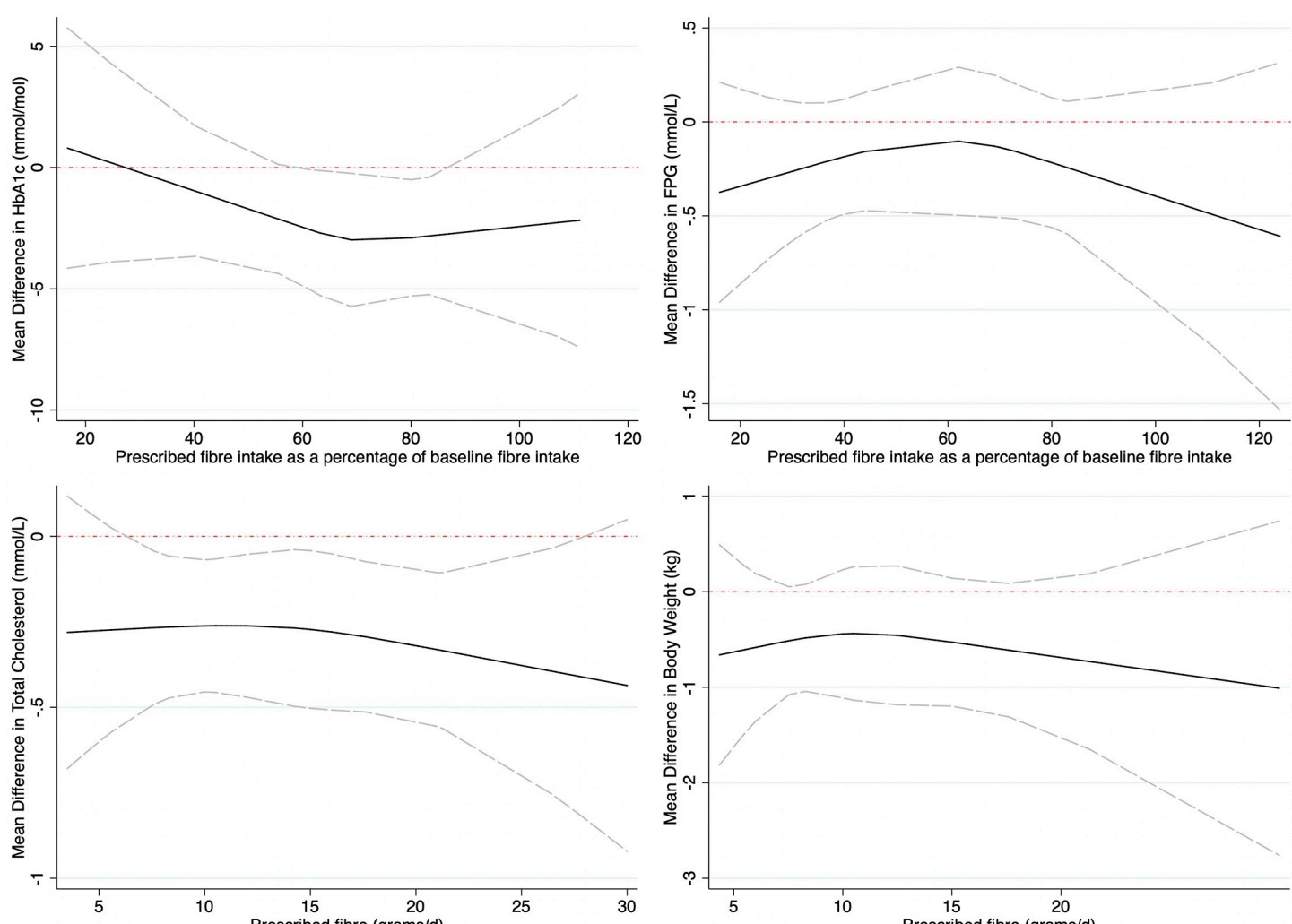

**Fig 4. Dose response curves of increasing fibre intakes and key cardiometabolic risk factors.** HbA1c (top left, data from 16 trials of 758 participants), fasting glucose (top right, data from 18 trials of 979 participants), total cholesterol (bottom left, data from 25 trials of 1,178 participants), and body weight (bottom right, data from 18 trials of 877 participants) from trials of increasing fibre intakes. The 95% CIs are shown as dashed lines. FPG, fasting plasma glucose.

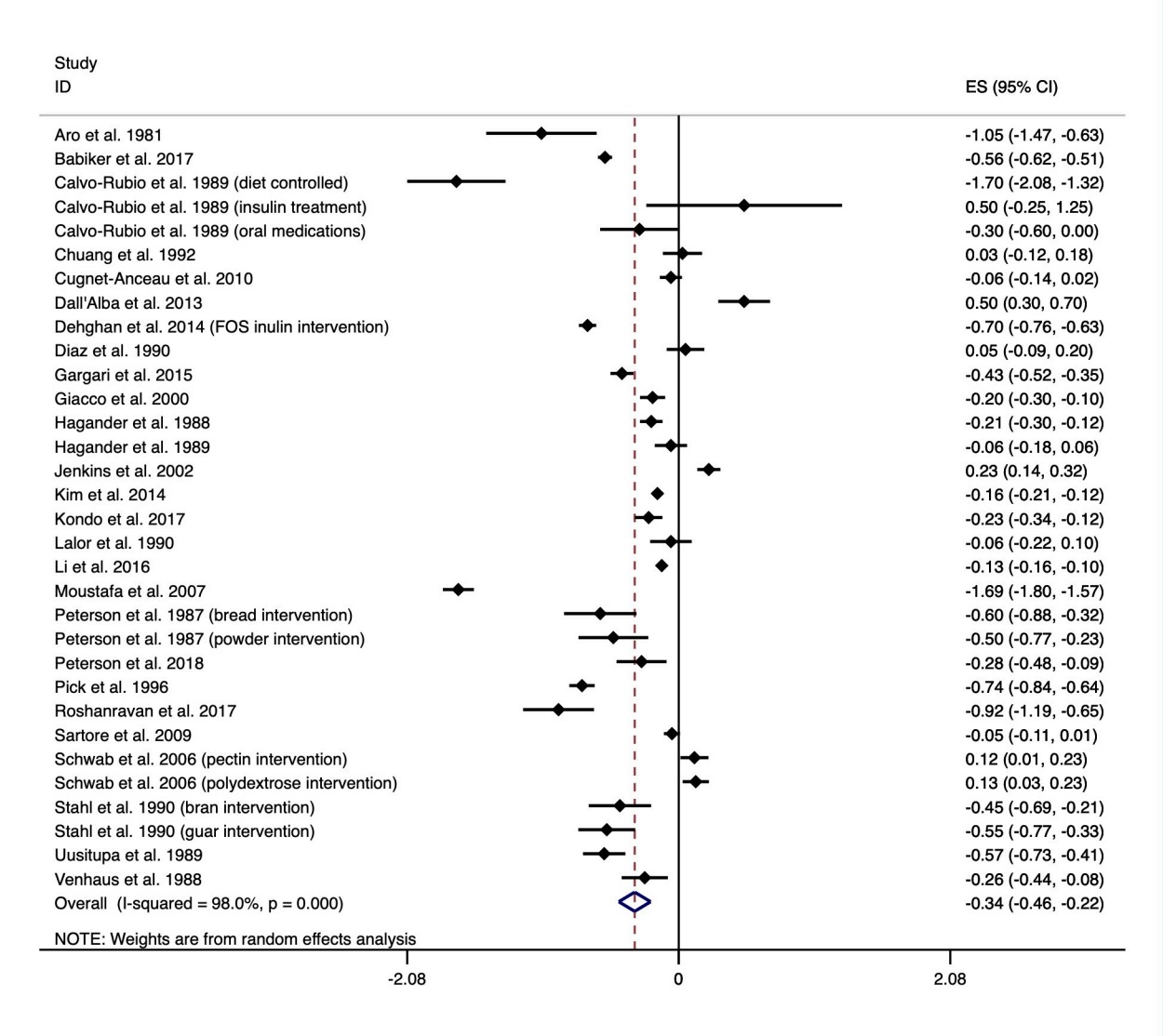

**Fig 5. Mean difference in total cholesterol (mmol/L) between intervention and control groups from trials of increasing fibre intakes.** Negative values show a decrease in total cholesterol when increasing fibre intakes (the intervention). ES, effect size; FOS, fructooligosaccharides.

data completeness and selective reporting, but overall assessment of bias was limited by insufficient reporting of sequence generation, allocation concealment, and blinding of participants and outcomes. Further information is contained in the supplementary materials (S2 Appendix).

When following GRADE protocols, evidence relating to all-cause mortality from prospective cohort studies was graded as moderate quality, having been upgraded because of a clear dose response. Evidence regarding fasting plasma glucose, total cholesterol, and LDL cholesterol were graded as high because of the dose response. Evidence from trials relating to most cardiometabolic risk factors was graded as being of moderate quality, having been downgraded for high heterogeneity. Full GRADE tables are shown in the supplementary material (S18 Appendix). The PRISMA checklist is shown in S19 Appendix.

## Discussion

### Summary of findings

The findings from our systematic review and meta-analyses have demonstrated the likely benefits of increasing dietary fibre in people with diabetes. Prospective cohort study data indicate a reduced risk of total mortality in adults with diabetes. Given that in many relatively affluent societies most adults consume around 20 g of dietary fibre per day [26], our data suggest that a 15 g increase to 35 g per day might be a reasonable target that would be expected to reduce risk of premature mortality by 10% to 48%. A similar finding to that observed for total mortality is apparent when considering cardiovascular mortality in relation to dietary fibre intakes, although confidence intervals were wide as a result of few events. Findings from controlled trials indicate that increasing intakes can improve glycaemic control, body weight, total and LDL cholesterol, and CRP, providing evidence to support the findings relating to total and cardiovascular mortality. Our analysis indicates these findings apply to those with type 1 and type 2 diabetes, whether treated on diet alone, oral agents, insulin, or a combination of treatments, as well as those with prediabetes.

### What the study adds to the existing literature

Our study extends the findings of a previous systematic review and meta-analysis of controlled trials that examined the effects of dietary fibre, regardless of source, in the management of diabetes [13]. That review considered only the effects on glycaemic control in people with type 2 diabetes. We were able to include 34 more trials that involved an additional 600 participants in both intervention and control groups, included people with type 1 or type 2 diabetes and prediabetes, and examined a range of additional outcomes. We excluded a number of trials included in the 2013 review, which involved changes in other nutrients in addition to an increase in dietary fibre [77,78], comparisons of one type of fibre with another [79–81], or compared a fibre-rich diet with another dietary intervention [82]. Two more recent reviews have been published; one considered the effects of viscous fibre [15], and the other was an umbrella review which provided no new analyses [16].

A substantial body of data derived from well-conducted controlled trials that have compared the effects on glycaemic control and cardiometabolic risk factors of relatively low (around 40%) and relatively high (around 60%) carbohydrate intakes in both type 1 and type 2 diabetes [7,8,10,11] support our findings and their clinical relevance. These studies were not included in our meta-analysis because our principal objective was to examine the effect of increasing dietary fibre intake without advice to change the overall carbohydrate load of the diet. Benefits, particularly in terms of glycaemic control and lipid profiles, were observed with higher carbohydrate intakes, but only when a high proportion of total carbohydrate was derived from foods rich in dietary fibre [11,12]. A high-carbohydrate–low-fibre diet was associated with similar or worse overall glycaemic control, lower HDL, and higher triglyceride levels when compared with lower carbohydrate intakes [83]. These findings were apparent even when energy balance was rigorously controlled. This appears to contradict our findings, which indicated that the effect of fibre on glycated haemoglobin was apparent only in trials that did not attempt to achieve energy balance, suggesting that weight loss may have been responsible for this effect of dietary fibre. However, we identified only 5 trials that measured glycated haemoglobin that involved weight control, so this observation may also have been because of the small number of trials.

### Strengths and limitations

The present study has a number of strengths. Arguably, the most important of these is the parallel consideration of the effects of higher fibre intakes in controlled trials and prospective

cohort studies. The former approach involved the examination of the effects on glycaemic control and cardiometabolic risk factors and the latter the extent to which these may translate into clinical outcomes. Although the analysis of only 2 cohort studies of participants with type 1 or type 2 diabetes may be seen as a limitation, participants were drawn from 22 countries. Additional information provided to us by the investigators of these studies enabled us to use country-specific effect size estimates. We were also able to demonstrate the dose response effect in terms of the relationship between dietary fibre and total mortality and that the relationship was not linear as had been previously been believed to be the case. Although the dose response effect between dietary fibre and cardiovascular mortality appeared similar to that observed for total mortality, the number of events was smaller and the upper confidence interval remained above 1. Nevertheless, these observations together with the relationship between dietary fibre and the cardiovascular risk factors observed in trials suggest that a dose response effect is indeed likely. So, although it is never possible to fully discount residual confounding in observational studies, the consistency of findings in the trials and cohort studies together with the dose response relationships suggest a causal association.

The meta-analysis of the intervention trials has potential limitations. All pooled analyses from controlled trials were subject to high heterogeneity, and therefore the strength of the evidence was, as required by GRADE, downgraded accordingly. However, such heterogeneity is to be expected given that trials included a variety of interventions and a range of dietary fibre intakes and were undertaken in a variety of populations consuming different diets and with different background lifestyle patterns. Our consideration of dietary fibre from supplements as well as from food sources, such as whole grains, may have contributed to the observed heterogeneity, however, not to the extent that it was detected by meta regression analyses for any outcome. Although there were only 2 controlled trials that were conducted in people with type 1 diabetes, meta regression analyses did not identify type of diabetes as a determining variable for any outcome, such as HbA1c. Furthermore Balk and colleagues [84] demonstrated a clear relationship between dietary fibre and HbA1c in the EURODIAB cohort of people with type 1 diabetes, in addition to the relationship between dietary fibre and total mortality shown by Schoenaker and colleagues [85]. Meta regression analyses also did not identify that potential sources of bias were determining variables of the pooled results, and sensitivity analyses did not identify data integrity issues for the majority of cardiometabolic outcomes. Instead, key variables beyond increasing fibre intake were the amount of fibre provided or consumed by participants at baseline, the global region where the trials were conducted, and participant inclusion criteria other than diabetes type. A further potential limitation to this work was the lack of prospective cohort data in non-European countries. However, the association between higher intakes of fibre and lower HbA1c has been observed in China [86] and Japan [87, 88]. The lack of long-term (12 months or greater) randomised controlled trials of increasing fibre intake in adults with diabetes is somewhat mitigated by the findings of the cohort studies. Thus, we believe that despite the observed heterogeneity the conduct of a meta-analysis was appropriate and our conclusions valid.

## Implications of our findings

Previous meta-analyses of clinical trials have demonstrated the potential of fibre supplements derived from psyllium [89] or viscous sources [15] to improve glycaemic control and cardiometabolic risk factors. European guidelines for the management of type 1 and type 2 diabetes have emphasised the benefits of soluble forms of dietary fibre from legumes [90]. However, our systematic review and meta-analyses that included trials in which fibre was increased either by the addition of supplements of extracted or synthetic fibres, or by the inclusion of

fibre rich foods, found that the source and type of dietary fibre did not influence the extent of improvements in glycaemic control or the various cardiometabolic measurements. When considering the findings of this research in relation to nutrition recommendations for the management of diabetes, it is noteworthy that the reduction in risk of premature mortality observed in the prospective cohort studies for both type 1 and type 2 diabetes was with fibre principally derived from food sources, rather than from manufactured foods to which extracted or synthetic fibre have been added. So, although the chemistry, physical properties, physiology, and metabolic effects of dietary fibre suggest that the observed benefits in terms of improvement in cardiometabolic risk factors should signal reduced morbidity and greater life expectancy in association with all types and sources of dietary fibre, the only direct evidence for such benefit derives from fibre as it occurs naturally in food. Thus, although a role for fibre supplements should not be excluded, the dose response data from the prospective cohorts studies supports the consumption of high fibre foods, which will most likely contain additional nutrients [91].

Despite not directly addressing the issue, these findings also have implications for recommendations relating to the total amount of carbohydrates in the diabetic dietary prescription. The case has been made for reductions in total carbohydrates, with suggestions that intake should be reduced to below 26% total energy (a 'low carbohydrate' diet) or to an even greater extent for a ketogenic diet [92]. Such suggestions have typically been based on relatively short-term studies (of up to 6 months duration), which have shown an improvement in glycaemic control and lipid profiles, especially in those with type 2 diabetes in which total carbohydrates has been reduced [93]. Such studies do not often consider whether the improvements observed are due to the lower carbohydrate load or the reduced energy intake. Longer term follow up indicates no lasting benefit in terms of cardiometabolic risk factors when low carbohydrate diets have been compared with more conventional dietary approaches [93–95].

## Conclusion

Aggregated data from intervention trials and cohort studies provide strong support for current nutrition recommendations [90,96], which advise that those with all types of diabetes should be encouraged to have adequate intake of dietary fibre. Vegetables, pulses, whole fruits, and whole grains are excellent sources. There is no suggestion from cohort studies or controlled trials that relatively high intake of these carbohydrate-rich foods are associated with deterioration of glycaemic control or weight gain. These findings do not detract from the widely accepted recommendation to reduce intakes of sugars and rapidly digested starches [97]. Our findings indicate that for those who choose a reduced intake of total carbohydrates, the inclusion of fibre supplements may provide the means of ensuring recommended intake. However, further long-term adequately powered studies will be required to establish whether fibre supplements will confer longer term clinical benefit that is comparable with fibre that occurs naturally in foods.

## Supporting information

**S1 Appendix. PICO tables: Fibre and whole grains in diabetes management.** PICO, Population, Intervention, Control, and Outcomes.
(DOCX)

**S2 Appendix. Identified studies.** Table A: Description of identified prospective studies. Table B: Description of identified intervention trials. Fig A: Cochrane risk of bias tool summary for the trials identified as eligible for this review.
(DOCX)

**S3 Appendix. Analyses for fibre and mortality.** Fig A: Higher versus lower analysis for all-cause mortality. Data from 6 European countries in EPIC and EURODIAB cohort with random effects model. Fig B: Higher versus lower analysis for cardiovascular mortality. Data from 6 European countries in EPIC and EURODIAB cohort with random effects model.
(DOCX)

**S4 Appendix. Analyses for fibre and HbA1c (mmol/mol).** Fig A: Mean difference in HbA1c (mmol/mol) between intervention and control groups from trials of increasing fibre intakes. Table A: Univariate meta regression analyses to test for interaction. Fig B: Dose response curve for HbA1c (mmol/mol) when increasing fibre intakes accounting for baseline value when the data were available.
(DOCX)

**S5 Appendix. Analyses for fibre and fasting plasma glucose (mmol/L).** Fig A: Mean difference in fasting glucose (mmol/L) between intervention and control groups from trials of increasing fibre intakes. Table A: Univariate meta regression analyses as tests for interaction. Fig B: Dose response curve for fasting plasma glucose (mmol/mol) when increasing fibre intakes accounting for baseline value when known.
(DOCX)

**S6 Appendix. Analyses for fibre and total cholesterol (mmol/L).** Fig A: Mean difference in total cholesterol (mmol/L) between intervention and control groups from trials of increasing fibre intakes. Table A: Univariate meta regression analyses as tests for interaction. Fig B: Dose response curve for total cholesterol (mmol/L) when increasing fibre intakes.
(DOCX)

**S7 Appendix. Analyses for fibre and LDL cholesterol (mmol/L).** Fig A: Mean difference in LDL cholesterol (mmol/L) between intervention and control groups from trials of increasing fibre intakes. Table A: Univariate meta regression analyses as tests for interaction. Fig B: Dose response curve for LDL cholesterol (mmol/L) when increasing fibre intakes. LDL, low-density lipoprotein.
(DOCX)

**S8 Appendix. Analyses for fibre and HDL cholesterol (mmol/L).** Fig A: Mean difference in HDL cholesterol (mmol/L) between intervention and control groups from trials of increasing fibre intakes. Table A: Univariate meta regression analyses as tests for interaction. Fig B: Dose response curve for HDL cholesterol (mmol/L) when increasing fibre intakes. HDL, high-density lipoprotein.
(DOCX)

**S9 Appendix. Analyses for fibre and triglycerides (mmol/L).** Fig A: Mean difference in triglycerides (mmol/L) between intervention and control groups from trials of increasing fibre intakes. Table A: Univariate meta regression analyses as tests for interaction. Fig B: Dose response curve for triglycerides (mmol/L) when increasing fibre intakes.
(DOCX)

**S10 Appendix. Analyses for fibre and body weight (kg).** Fig A: Mean difference in body weight (kg) between intervention and control groups from trials of increasing fibre intakes. Table A: Univariate meta regression analyses as tests for interaction. Fig B: Dose response curve for body weight (kg) when increasing fibre intakes.
(DOCX)

**S11 Appendix. Analyses for fibre and BMI (kg/m$^2$).** Fig A: Mean difference in BMI (kg/m$^2$) between intervention and control groups from trials of increasing fibre intakes. Table A: Univariate meta regression analyses as tests for interaction. BMI, Body Mass Index.
(DOCX)

**S12 Appendix. Analyses for fibre and waist circumference (cm).** Fig A: Mean difference in waist circumference (cm) between intervention and control groups from trials of increasing fibre intakes. Table A: Univariate meta regression analyses as tests for interaction.
(DOCX)

**S13 Appendix. Analyses for fibre and systolic blood pressure (mmHg).** Fig A: Mean difference in systolic blood pressure (mmHg) between intervention and control groups from trials of increasing fibre intakes. Table A: Univariate meta regression analyses as tests for interaction. Fig B: Dose response curve for systolic blood pressure (mmHg) when increasing fibre intakes.
(DOCX)

**S14 Appendix. Analyses for fibre and diastolic blood pressure (mmHg).** Fig A: Mean difference in diastolic blood pressure (mmHg) between intervention and control groups from trials of increasing fibre intakes. Table A: Univariate meta regression analyses as tests for interaction. Fig B: Dose response curve for diastolic blood pressure (mmHg) when increasing fibre intakes.
(DOCX)

**S15 Appendix. Analyses for fibre and C-reactive protein (SMD).** Fig A: Standardised mean difference in CRP between intervention and control groups from trials of increasing fibre intakes. Table A: Univariate meta regression analyses as tests for interaction. CRP, C-reactive protein; SMD, standardised mean difference.
(DOCX)

**S16 Appendix. Analyses for fibre and fasting plasma insulin (SMD).** Fig A: Standardised mean difference in fasting insulin between intervention and control groups from trials of increasing fibre intakes. Table A: Univariate meta regression analyses as tests for interaction. SMD, standardised mean difference.
(DOCX)

**S17 Appendix. Analyses for fibre and HOMA IR (mg/dL).** Fig A: Mean difference in HOMA IR (mg/dL) between intervention and control groups from trials of increasing fibre intakes. Table A: Univariate meta regression analyses as tests for interaction. Fig B: Dose response curve for HOMA IR (mg/dL) when increasing fibre intakes. HOMA IR, homeostatic model assessment of insulin resistance.
(DOCX)

**S18 Appendix. GRADE tables.** Table A: Do greater intakes of total dietary fibre reduce the risk of all-cause and CVD mortality for adults with type1 or type 2 diabetes? Table B: What is the effect of increasing dietary fibre intakes on HbA1c (mmol/mol) in diabetes management? Table C: What is the effect of increasing dietary fibre intakes on fasting plasma glucose (mmol/L) in diabetes management? Table D: What is the effect of increasing dietary fibre intakes on total cholesterol (mmol/L) in diabetes management? Table E: What is the effect of increasing dietary fibre intakes on LDL cholesterol (mmol/L) in diabetes management? Table F: What is the effect of increasing dietary fibre intakes on HDL cholesterol (mmol/L) in diabetes management? Table G: What is the effect of increasing dietary fibre intakes on triglycerides (mmol/L) in diabetes management? Table H: What is the effect of increasing dietary fibre intakes on body weight (kg) in diabetes management? Table I: What is the effect of increasing dietary

fibre intakes on BMI in diabetes management? Table J: What is the effect of increasing dietary fibre intakes on waist circumference (cm) in diabetes management? Table K: What is the effect of increasing dietary fibre intakes on fasting plasma insulin (standardised mean difference) in diabetes management? Table L: What is the effect of increasing dietary fibre intakes on HOMA IR (mg/dL) in diabetes management? Table M: What is the effect of increasing dietary fibre intakes on systolic blood pressure (mmHg) in diabetes management? Table N: What is the effect of increasing dietary fibre intakes on diastolic blood pressure (mmHg) in diabetes management? Table O: What is the effect of increasing dietary fibre intakes on C-reactive protein in diabetes management? BMI, Body Mass Index; CVD, Cardiovascular disease; GRADE, Grading of Recommendations Assessment, Development and Evaluation; HDL, high-denisty lipoprotein; HOMA IR, homeostatic model assessment of insulin resistance; LDL, low-density lipoprotein.
(DOCX)

**S19 Appendix. PRISMA Checklist for reporting of systematic reviews.**
(DOCX)

## Acknowledgments

We warmly acknowledge the authors who responded to our requests for further clarification or additional analyses, in particular, Associate Professor Joline Beulens, Associate Professor Sabita Soedamah-Muthu, Dr. Danielle Schoenaker, and Professor Monika Toeller. We also thank our translators for their work on publications in languages other than English: Dr. Alexandra Chisholm, Ms. Willemijn de Bruin, Dr. Zheng Feei Ma, Ms. Minako Kataoka, Dr. Luciano Rigano, Dr. Santiago Rodríguez-Jiménez, Dr. Pouya Saeedi, and Dr. Olya Shatova.

## Author Contributions

**Conceptualization:** Andrew N. Reynolds, Ashley P. Akerman, Jim Mann.

**Data curation:** Andrew N. Reynolds, Ashley P. Akerman.

**Formal analysis:** Andrew N. Reynolds, Ashley P. Akerman.

**Investigation:** Andrew N. Reynolds, Ashley P. Akerman, Jim Mann.

**Methodology:** Andrew N. Reynolds, Ashley P. Akerman, Jim Mann.

**Project administration:** Andrew N. Reynolds.

**Resources:** Andrew N. Reynolds.

**Validation:** Andrew N. Reynolds, Ashley P. Akerman.

**Visualization:** Andrew N. Reynolds, Ashley P. Akerman, Jim Mann.

**Writing – original draft:** Andrew N. Reynolds, Jim Mann.

**Writing – review & editing:** Andrew N. Reynolds, Ashley P. Akerman, Jim Mann.

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
