## [Decision Letter · Decision Letter 0]

25 Sep 2019

Dear Dr. Reynolds,

Thank you very much for submitting your manuscript "Dietary fibre in diabetes management: systematic review and meta analyses" (PMEDICINE-D-19-01995) for consideration at PLOS Medicine. 

[LINK]

In light of these reviews, I am afraid that we will not be able to accept the manuscript for publication in the journal in its current form, but we would like to consider a revised version that addresses the reviewers' and editors' comments. Obviously we cannot make any decision about publication until we have seen the revised manuscript and your response, and we plan to seek re-review by one or more of the reviewers. 

We expect to receive your revised manuscript by Oct 09 2019 11:59PM. Please email us (plosmedicine@plos.org) if you have any questions or concerns.

We look forward to receiving your revised manuscript. 

Sincerely,

Adya Misra, PhD

Senior Editor 

PLOS Medicine

plosmedicine.org

Abstract please combine methods and findings into one section

Abstract-last sentence of the methods and findings section should include the limitations of your methodology 

Abstract- last sentence “… reduce risk of premature mortality”. Should this be followed by…” in diabetic patients?”

Please include an “Introduction” heading at the start of the introduction

Line 70- please remove reference to “new” or clarify if this is an update to a previously published systematic review. Please provide a citation for the same. 

Line 72- “has” not “have”

Line 77-78 please avoid assertions of primacy 

Page 7- please move the role of the funding source to the “Financial disclosure” section 

Please provide p values along with 95% confidence intervals where appropriate 

Please present and organize the Discussion as follows: a short, clear summary of the article's findings; what the study adds to existing research and where and why the results may differ from previous research; strengths and limitations of the study; implications and next steps for research, clinical practice, and/or public policy; one-paragraph conclusion.

Line 277- a systematic review cannot demonstrate any health benefit, please revise as necessary

Line 280-281-please clarify if the reduced risk of mortality is for the general population or for patients with diabetes 

Please remove page numbers from the PRISMA checklist as these are likely to change 

Comments from the reviewers:

Reviewer #1: This manuscript presents results of a systematic review and meta-analysis of dietary fiber in diabetes management. The rationale for conducting the study is well-developed. The authors included 2 prospective studies and 41 intervention trials, and demonstrated the benefits of increasing dietary fiber for reducing mortality and a series of cardiometabolic risk factors in prediabetes, type 1 and 2 diabetes. The statistical analysis is adequate and the potential sources of bias are well considered for each outcome. Here are some major concerns about the study:

1. The number of prospective studies may be too small for a meta-analysis. Therefore, the authors may want to focus on clinical trials (I found one previous publication in Plos Medicine might be informative. PMID: 27434027), and provide some comments two prospective studies, In addition, causal inferences (such as "Cardiovascular mortality risk reduction" in line 169) can rarely be made from observational studies, and more conservative description would be preferred. If the authors want to keep meta-analysis of prospective studies, then influence tests are strongly discouraged as there are only two studies.

2. Some believes that a meta-analysis should not be performed when between-study heterogeneities are large(as in this study), because it shows that findings from these studies are essentially different and cannot be solely explained by random errors. The authors may want to defense their decision to perform a meta.

3. The generalization of conclusion to type 1 diabetes has been listed as a strength of while only two studies have been conduced among T1D patients. The authors may want to comment on if dietary advices for T1D and T2D management are interchangeable, and why an analysis stratified by diabetes type was not performed.

4. Except for mortality and glycemic control, 11 additional cardiometabolic risk factors were discussed in this study. The results of dose response analysis for each outcome in the manuscript text were so brief that need to be further discussed.

Minor concerns:

Line 75, might be useful to mention which CVD risk factors have been considered.

Line 76, "and whether these benefits translate into a reduction in morbidity and mortality" How has this question been addressed ? Does this imply the inclusion of prospective cohort studies ?

Line 175, "This resulted in an absolute risk reduction of 14 (95%CI 4 to" how was this data derived ?

Line 185, it might be useful to describe some study characteristics briefly, such as sample size, doses, % men/women, disease status(T1D/T2D), and duration of intervention.

The flow chart (Figure 1) is confusing-- out of 8,050 records retrieved from initial search, however, the sum of the records from different databases below is 8,086; 42 studies were included according to the flow chart while there are 43 identified studies in Table S1 and Table S2.

Reviewer #2: See attachment

Michael Dewey

Reviewer #3: The systematic review and meta-analysis from Reynolds et al titled: "Dietary fibre in diabetes management: systematic review and meta analysis" provides relevant and comprehensive update on previous literature in this area on a wide scope of outcomes. However, there are several concerns that need to be addressed, outlined below:

- In the introduction, the authors rationalize the need for the systematic review and meta analysis through dietary CHO focus, citing that (lines 63-65) controlled trials confirm potential of relatively high to low CHO intakes in diabetes. The latest SRMA however, does not fully support this stance. Moreover, the SRMA cited from 2013 refer mostly to the benefit of fiber addition/supplementation rather than as part of a high CHO diet. The authors should provide some rationale for the trial in the context of more recent research/meta analysis on whole grain/dietary fibre in DM management (McRae MP. J Chirop Med. 2018; Wang Y et al. Int J Mol Epid Genet. 2019; Jovanovski E et al, Diabetes Care, 2019,e tc ). 

- With regards to analysis of mean difference (MD) between intervention and control groups (line 131), please indicate whether MD between end values or MD between change-from-baseline values were used and whether there was an order of preference. I.e. order of preference could be use of change values provided by article if available, followed by calculation of change values where appropriate, followed by end values provided by article and lastly calculation of end values. 

- Please provide detail on how the trials with multiple comparison arms were treated? If compared to a single control arm, were the comparisons combined to create a single pair-wise comparison or were they included as separate comparisons? If the latter, were there any adjustments made (ie. was the SE(MD) adjusted) to take this duplicate comparison into account? I.e. In the forest plot for HbA1c (figure 3) it appears that multiple intervention arms from the same trial were listed separately. 

- In line 144, please elaborate on the influence analysis, i.e. what determined whether a study was influential? 

- Concern that the search is not comprehensive. The authors have used only "dietary fiber/re" or whole grain terms to try to 'comprehensively' capture all research on dietary fiber. It is likely that many possibly eligible trials were missed by not including explicit food search terms or specific known fibers/whole grains or their sources (ie for beans (legumes), oats, cereal etc)... It appears that trials fitting selection criteria have been missed (ie McGeoch et al., 2013 etc). The review here included many cardiometabolic outcomes, such as blood lipids, blood pressure, etc. However, the presented online search term only included glycemia-related search for outcomes (Search#2 Supplementary, pg 3). Again, it is possible that some trials relating to these outcomes may have been missed. 

- Within the dose response analysis, how did the authors collectively assess doses of whole grain sources ie 50g of barley, containing ~8% b-glucan fiber, relative to more isolated sources of dietary fiber? Did the authors extrapolate/estimate the amount of fiber from whole food sources? The dose response analysis would otherwise not have much merit. In S1 under a priori subgroup analysis: Amount: is the volume of fibre or whole grain used? 

- A priori subgroup analysis included "fibre type: all fibre or just one type (i.e inulin) viscosity, solubility". Please clarify in more detail what fibre types were considered, which 'viscosity' and 'solubility. Is this an inclusive list of all the subanalysis that were performed? 

- In the GRADE assessment authors report on categorical analysis of study participant health status (ie presence/absence of renal issues etc). This was not included in the PICO table of a priori subgroup analysis? Please indicate exactly which variables were tested, and whether these variables were determined apriori. 

- In table S3 authors exclude based on HbA1c, aged over 65 etc. Is this categorically assessed (ie over vs uder 65?). What does exclusion based on HbA1c or BMI mean? Please be more concise in provision of explanations. 

- What singular fibre types were assessed as a dicotomous variables (is Table S1)?

- Were there any restrictions on study duration? It is questionable whether short term trials (ie <3-4wks) are relevant on certain outcomes, especially on your primary outcome. 

- The dose response curve for HbA1c is presented as % intake from baseline. How much data was available? The authors do not report on the dose response relationship per amount of fibre intake per se. Does the relationship still hold? 

- We need to be very careful in conclusions that the study results cannot be generalized to whole grain as only very few studies evaluated a whole grain. Perhaps that identifies future areas of research. Most of the data is on dietary fiber supplements and concentrates and this perhaps should be acknowledged. 

- Please add a section on study limitations. Some limitations are already mentioned in discussion.

- Discussion: Lines 283-284. Was this statement based on subanalysis of background treatment type (ie diet alone vs oral agents, vs insulin vs combination) for each of the 5 outcomes?

- Discussion: Line 324-325. Please provide data on MD according to each category (T1DM, T2DM, pre-diabetes). There are very few studies in the non-T2DM categories. 

- Discussion: Line 346: The authors mention previous meta analysis of RCTs demonstrating potential of psyllium and viscous fibers to affect glycemic control. These SRMAs report on significantly larger magnitudes of change (ie the latter SRMA, the effect of viscous fiber is approx. 3 fold higher, including both extracted fibers and whole foods -ie oats). It would be useful to comment on the difference observed and potential of different fiber types to improve magnitude of effect of fiber, rather than focus on all fiber that is highly heterogeneous (as demonstrated here). 

- Lines 356-360. Similarly, we need to be careful when extrapolating conclusions on cardiometabolic risk (as the 'only direct evidence for such benefit") based on 2 cohorts that have inherent risk of confounding and in this instance, have not shown a cardiovascular benefit. While the authors rightly conclude that the consumption of high fibre foods should be encouraged (line 361), the conclusions should be in the context of the current evidence from RCTs and fiber sources they studied. 

- The authors comment in the discussion on reduced risk of total mortality, but not on results of CV mortality which is the focus of the assessed RF in this review. 

- Discussion: Lines 293-304 and 363-372 further discuss comparison of low/high carbohydrate diets which are not considered or explored in this SRMA (as sub-analysis of background diet) and thus may not be very relevant in this context. The fiber in majority of trials is supplemented to the diet. These sections should be less elaborate. 

- Lines 305-305: The authors suggest that the weight-loss may be 'principally responsible' for the observed effect on HbA1c. However, the MD in weight-loss reported here and in other SRMAs of dietary fibre is minimal (MD of <1kg) and likely not sufficient to generate the HbA1c reduction observed in this and other SRMAs. 

- Minor: Figure 3 describing mean difference in HbA1c contains random descriptors of study (ie either dose, type of fiber, diet in brackets or no descriptor, without apparent systematical order. Please revise.

- Minor: Line 138: Prescribed or measured fibre increase was used in dose response testing. Which was taken when both are reported?

[LINK]

---

## [Decision Letter · Decision Letter 1]

26 Nov 2019

Dear Dr. Reynolds,

Thank you very much for submitting your manuscript "Dietary fibre in diabetes management: systematic review and meta analyses" (PMEDICINE-D-19-01995R1) for consideration at PLOS Medicine. 

[LINK]

In light of these reviews, I am afraid that we will not be able to accept the manuscript for publication in the journal in its current form, but we would like to consider a revised version that addresses the reviewers' and editors' comments. Obviously we cannot make any decision about publication until we have seen the revised manuscript and your response, and we plan to seek re-review by one or more of the reviewers. 

We expect to receive your revised manuscript by Dec 17 2019 11:59PM. Please email us (plosmedicine@plos.org) if you have any questions or concerns.

We look forward to receiving your revised manuscript. 

Sincerely,

Adya Misra, PhD

Senior Editor 

PLOS Medicine

plosmedicine.org

Title- in accordance with comments from Ref 2- please amend title to “Whole grain fibre and management of diabetes: a systematic review and meta-analysis”

Please include a space between text and reference brackets 

Page 6-please avoid assertions of primacy 

Discussion- a systematic review cannot demonstrate any health benefit, please revise as necessary. Please also tone down conclusions to avoid overreaching. 

Please update your literature search, in accordance with comments from Reviewer 3

Comments from the reviewers:

Reviewer #2: The authors have addressed my points.

I still feel that prediction intervals are helpful in the presence of heterogeneity because they tell us about what might be the result from a future study. By contrast the confidence intervals tell us about the precision of the overall mean. If on reflection the authors still feel they are not informative I would not want to push the issue.

Michael Dewey

Reviewer #3: The authors have addressed some of the raised issues in sufficient manner. However, there are several concerns that remain which are very relevant and should be carefully considered before proceeding:

A concern remains that the search is not systematic and comprehensive. The authors state that food sources other than whole grain were not included as search terms. This limits the review to comprehensively searching only whole grain types of fiber. The authors should make this more prominent in their conclusions. The reader should not be misguided to infer all dietary fiber was included. 

Similarly, there are a number of cardiometabolic outcomes reported from trials in addition to glycemic measures. The search terms do not include these. While the authors suggest hand search was employed and they believe they would detect most through the glycemic search terms, this is not a systematic review. This is an important limitation. 

The last search was done in Jan 2019. This approaches one year (from publication time). A search update is strongly recommended. 

The authors state that all dose response testing used the fiber dose. In many trials, the fiber dose is not reported (ie when oats/whole grain is administered). If not available (ie directly from authors), how was the fiber dose estimated (ie of b-glucan in whole oats or barley, psyllium fiber)? 

A priori subgroup analysis included type of fiber, viscosity and solubility. The authors did not report on which fibers were categorized in these groups in order to run sub-analysis. Please clarify.

The overall tone and discussion of paper puts emphasis on whole grain. However, majority of trials appear to be supplementation of fiber from various sources to diet (and 2 cohorts). While the authors argue that meta regression did not identify differences between 2 sources (p=0.057), we must be careful how we interpret the available evidence. Evidence, based on the present search for whole grain, is still scarce (only n=4 trials for HbA1c), and most of the data refers to supplemented type. The conclusions therefore need to be cautionary. 

Similarly, the concern remains that strong recommendations and conclusions in discussion are based on only 2 European cohorts (ie "the only direct evidence for benefit derives from fibre as it occurs naturally in food), whereas most of the evidence from controlled trials on cardiometabolic risk is on fibre supplement (if discussing natural fiber from foods, ideally all food sources including legumes should have been included in this review). 

Despite that meta regression did not identify differences in effect size based on diabetes type, etc. for transparency purposes, information on MD + CIs for each subgroup estimate is warranted, especially in light of multiple comments in this regard.

The authors continue to suggest that the weight-loss observed here may be 'principally responsible' for the observed effect on HbA1c. The MD in body weight was -0.6kg. Guidelines generally cite that sustained weight loss of ≥5kg may improve glycemic control. If deciding to keep, please provide references for the statement in the response referring to <1kg change in weight. 

The authors recommend that in those with prediabetes, type 1, or type 2 diabetes - should increase their dietary fibre intakes by 15 grams per day or to 35 grams per day. Does this hold for each diabetes category? What is the basis for 15g? The recommendations should be more cautionary (very little data for type1 DM exists). 

Minor note: Sartore 2009 is not randomized trial. Please modify the Table of Characteristics.

Minor note: It would be useful to provide the data for HbA1c in % (NGSP), perhaps in brackets for ease of reader.

[LINK]

---

## [Editor Report · Decision Letter 2]

22 Jan 2020

Dear Dr. Reynolds,

Thank you very much for re-submitting your manuscript "Dietary fibre in diabetes management: systematic review and meta analyses" (PMEDICINE-D-19-01995R2) for review by PLOS Medicine.

I am pleased to say that provided the remaining editorial and production issues are dealt with we are planning to accept the paper for publication in the journal.

[LINK]

In revising the manuscript for further consideration here, please ensure you address the specific points made by each reviewer and the editors. Please also check the guidelines for revised papers at http://journals.plos.org/plosmedicine/s/revising-your-manuscript for any that apply to your paper. In your rebuttal letter you should indicate your response to the reviewers' and editors' comments and the changes you have made in the manuscript. Please submit a clean version of the paper as the main article file. A version with changes marked must also be uploaded as a marked up manuscript file.

We look forward to receiving the revised manuscript by %Jan 29 2020 11:59PM. 

Sincerely,

Adya Misra, PhD

Senior Editor 

PLOS Medicine

plosmedicine.org

Requests from Editors:

Author summary- could you provide a sentence in “why was the study done” to outline the importance of fibre in the diet

Author summary- I believe Line 63 has an additional “at”

Title- Please alter the title to include grains in line with comments from Ref 3 in the last round

Methods- please specify, if that’s the case that PRISMA reporting guidelines were used. In addition please provide a reference to the completed checklist 

Methods- you have not mentioned a search update as previously discussed. Please note that we are coming up to a year since the last search date and it is standard practice to update the search to ensure no new articles are missed 

Methods- please provide information about quality assessment as done in the abstract

Throughout- please ensure all p values are “p” rather than P. Please also revise “HbA1c” to include c in lower case and not subscript 

Figure 2 requires clarification. For instance- does this include Type 1 or 2 diabetes patients

Figure 4 requires clarification. For instance please provide a key to the graphs and explain which cardiovascular factors were included

Discussion- In line with the comments from Ref 3 in the last round, in the limitations section please provide a brief explanation regarding the inclusion of grains/legumes in the current meta-analysis and what impact this has on the conclusions (if any). 

Methods- please indicate that a full search strategy has been provided as SI 

PRISMA checklist- please remove page numbers as these are likely to change and use paragraphs or sections instead 

Comments from Reviewers:

[LINK]

---

## [Editor Report · Decision Letter 3]

3 Feb 2020

Dear Dr. Reynolds, 

On behalf of my colleagues and the academic editor, Dr. Ronald Ma, I am delighted to inform you that your manuscript entitled "Dietary fibre and whole grains in diabetes management: systematic review and meta analyses" (PMEDICINE-D-19-01995R3) has been accepted for publication in PLOS Medicine. 

PRODUCTION PROCESS

PRESS

PROFILE INFORMATION

Thank you again for submitting the manuscript to PLOS Medicine. We look forward to publishing it. 

Best wishes, 

Adya Misra, PhD

Senior Editor 

PLOS Medicine

plosmedicine.org